# Feasibility study for supporting medication adherence for adults with cystic fibrosis: mixed-methods process evaluation

Daniel Hind [ID],[1] Sarah J Drabble,[2] Madelynne A Arden,[3] Laura Mandefield,[4] Simon Waterhouse,[1] Chin Maguire,[1] Hannah Cantrill,[1] Louisa Robinson,[1] Daniel Beever,[1] Alex Scott [ID],[1] Sam Keating,[1] Marlene Hutchings,[5] Judy Bradley,[6] Julia Nightingale,[7] Mark I Allenby,[7] Jane Dewar,[8] Pauline Whelan,[9] John Ainsworth,[9] Stephen J Walters [ID],[2] Martin J Wildman,[2,5] Alicia O'Cathain [ID] [2]

► Prepublication history and supplemental material for this paper is available online. To view these files, please visit the journal online (http://dx.doi.org/10.1136/bmjopen-2020-039089).

**Correspondence to**
Dr Daniel Hind;
d.hind@sheffield.ac.uk

## ABSTRACT

**Objectives** To undertake a process evaluation of an adherence support intervention for people with cystic fibrosis (PWCF), to assess its feasibility and acceptability.

**Setting** Two UK cystic fibrosis (CF) units.

**Participants** Fourteen adult PWCF; three professionals delivering adherence support ('interventionists'); five multidisciplinary CF team members.

**Interventions** Nebuliser with data recording and transfer capability, linked to a software platform, and strategies to support adherence to nebulised treatments facilitated by interventionists over 5 months (± 1 month).

**Primary and secondary measures** Feasibility and acceptability of the intervention, assessed through semistructured interviews, questionnaires, fidelity assessments and click analytics.

**Results** Interventionists were complimentary about the intervention and training. Key barriers to intervention feasibility and acceptability were identified. Interventionists had difficulty finding clinic space and time in normal working hours to conduct review visits. As a result, fewer than expected intervention visits were conducted and interviews indicated this may explain low adherence in some intervention arm participants. Adherence levels appeared to be >100% for some patients, due to inaccurate prescription data, particularly in patients with complex treatment regimens. Flatlines in adherence data at the start of the study were linked to device connectivity problems. Content and delivery quality fidelity were 100% and 60%–92%, respectively, indicating that interventionists needed to focus more on intervention 'active ingredients' during sessions.

**Conclusions** The process evaluation led to 14 key changes to intervention procedures to overcome barriers to intervention success. With the identified changes, it is feasible and acceptable to support medication adherence with this intervention.

**Trial registration number** ISRCTN13076797; Results.

### Strengths and limitations of this study

► This is a detailed evaluation of the feasibility of an adherence support system for people with cystic fibrosis.
► The use of mixed-methods provided indepth understanding of the processes involved in delivering the service, its value and factors that might influence its use, implementation and success.
► This was a small, two-centre study.

## BACKGROUND

Cystic fibrosis (CF) is a life-threatening, inherited condition affecting over 90 000 people worldwide, primarily of Northern European ancestry.[1] Median survival for people with cysticfibrosis (PWCF) is estimated at 31 years[2–6] with progressive lung function decline, caused by regular infection and damage to airways, being one of the main disease features.[2]

Preventative medications preserve lung function and reduce exacerbations.[7–13] Low adherence to these medications is problematic as this predicts exacerbations requiring intravenous antibiotics (IVAB).[14 15] Exacerbations of this nature carry a risk of systemic side effects of both increased mortality,[16 17] and cost of care.[18–20] In 2012, the total spend on CF in the UK was estimated to be £100 million, with £30 million spent on inhaled antibiotics and mucolytics[21]; the UK CF population received 1 71 907 days of IVAB with 93 455 days received in hospital, costing an estimated £27 million.[22]

Self-reported adherence to inhaled therapies underestimates objectively measured adherence, with rates of 80% and 36% recorded, respectively[23] and systematic data

collection suggests objective adherence to be closer to 30%.[24] As a result, clinicians are currently unable to identify PWCF with low adherence, in order to provide additional support. Hitherto, the most objective surrogate measure of adherence has been the medicines possession ratio (MPR). However, based on the experience of a CF service in Leeds, UK, MPR rates of 63%[25] considerably overestimate adherence compared with nebuliser download data of 36%.[26]

Treatment burden has long been recognised as a key barrier to medication adherence in CF,[27] and reducing treatment burden is a key research priority for PWCF and clinicians, identified by the Cystic Fibrosis Foundation and the James Lind Alliance.[28 29] In response, a complex intervention was developed to support inhaled medication adherence in PWCF.[30] This article presents the results of a process evaluation that was undertaken alongside a pilot randomisedcontrolled trial (RCT), the objectives of which were to determine the feasibility of a full-scale RCT.[30] Here, we describe the resultant changes made to intervention procedures prior to that full-scale RCT.[31] The specific objectives of the process evaluation were:

1. To triangulate qualitative and quantitative data collected on intervention inputs, engagement, activities and contextual factors, alongside immediate and intermediate outcomes recorded in the feasibility study, to understand and identify potential barriers to intervention implementation and success.
2. To document and use these findings to guide changes to intervention procedures, ahead of a future, full-scale RCT.

## METHODS
### The wider feasibility study
The process evaluation forms one part of a wider pilot study, which also assessed the feasibility of RCT procedures and mechanisms of action (reported elsewhere[30 32]). The pilot RCT consisted of 33 intervention patients and 31 control patients. Three trained interventionists in two UK CF centres delivered the intervention to PWCF in the intervention arm and followed them up for 5 months, ±1 month.

### Intervention description
The complex intervention to support adherence in CF was developed to enable PWCF to manage adherence to nebulised medication, with a view to shifting CF treatment from rescue in hospital settings to prevention, managed in the community. The full intervention development process is described in a separate article.[30]

The complex intervention consists of four key elements: the eTrack, CFHealthHub (CFHH) server, the CFHH

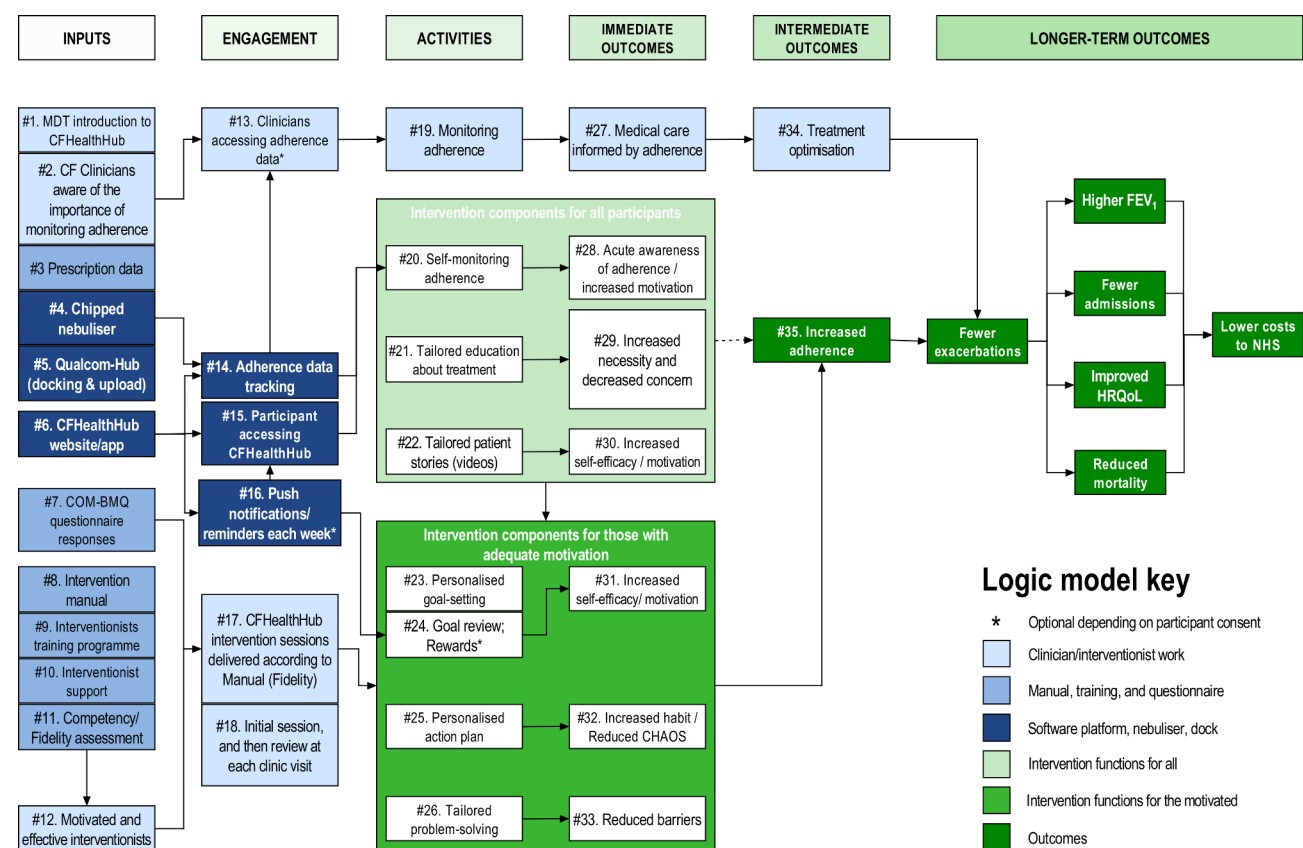

**Figure 1** Logic model. CF, cysticfibrosis; CHAOS, Confusion, Hubbub and Order Scale; COM-BMQ, Capability Opportunity Motivation Behaviour Beliefs about Medicines Questionnaire; FEV$_1$,forced expiratory volume in 1 second; HRQoL, Health-Related Quality of Life; MDT, multidisciplinary team.

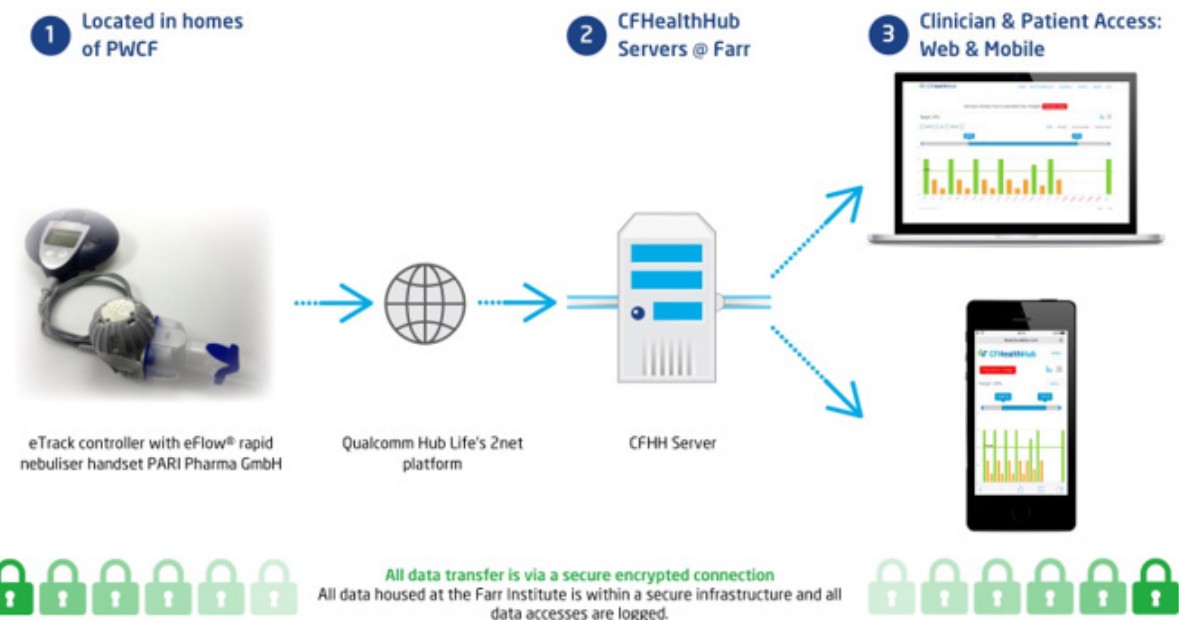

**Figure 2** The digital platform. CFHH, CFHealthHub; PWCF, peoplewith cystic fibrosis.

Apps and the manualised behavioural intervention. A logic model (figure 1) was produced to reflect, in detail, constructs and processes by which the intervention was expected to function; this is in terms of inputs, engagement, activities and outcomes. The logic model's hashed numbers (#1, #2, etc) provide a reference for linking intervention materials and processes to logic model constructs in figure 1.

The eTrack (PARI Pharma GmbH, Starnberg, Germany; #4) is a microchipped nebuliser, enabling real-time monitoring of adherence to nebulised medications. Time-stamped records of medications administered via the eTrack are sent to a 2net Hub (Qualcomm, San Diego, USA; #5) which transmits data to PARI.

Real-time inhalation data are received by the CFHH server infrastructure, stored securely and used for display in both a web-based interface and a mobile app (#6, see figure 2). Each of these displays adherence data alongside tools to support behaviour change and educational content.[33] Educational modules within CFHH include: 'What is Cystic Fibrosis?'; 'What does my IV treatment do?'; 'I'm not convinced that my nebuliser treatment works'; 'What does my nebuliser treatment do and why should I take it?'; 'Why is it important that I do my nebuliser treatment every day?'; and, 'I have concerns about my nebuliser treatments'. The nebuliser medication information displayed to the user in these sections are tailored to them based on a baseline assessment of motivation, so as not to overwhelm them.

Participants and their interventionists had access to adherence displays for monitoring (#13, #19, #20) and other CFHH content (#21–#26), such as education about treatments (#21) and problem solving in the face of adherence barriers (#26). Interventionists would use CFHH to facilitate delivery of manualised behavioural intervention sessions (#8, #17).

Interventionists (n=3) included a clinical psychologist, a physiotherapist and a social worker. They received specific training to deliver the manualised intervention sessions (#9). Training was delivered over 2 days, in face-to-face workshops. This was supplemented by online learning modules and a further 4-week training schedule. Interventionists were assessed with online theory tests and in a competency assessment which examined intervention delivery within the first five sessions.

Sessions were delivered either face-to-face or remotely, on a one-to-one basis. All intervention arm participants received an initial intervention visit and a minimum of one additional review visit over the period of the study (#18). The content of sessions varied by participant reported motivation; sessions for those with low motivation were tailored to promote relationship/confidence building and to support the participant in the exploration of relevant CFHH educational and information material (#21, #22). Relevant material could be added to the participant's personalised 'Toolkit'. Sessions conducted with participants displaying higher motivation would also involve supporting the participant to set personalised adherence goals (#23, #24), and to make action plans (#25) and engage in problem solving including making coping plans where relevant (#26).

## Design

A mixed-methods approach was used for the process evaluation. Although this pragmatic case study[34][35] primarily works at the level of the programme, we also present a nested multiple case design, with cases at the level of the PWCF, and two embedded units of analysis—interviews with intervention participants and trial data.

## Data sources

Quantitative and qualitative data sources were triangulated to address process evaluation objectives. These are described using hashed numbers to relate data sources to aspects of the logic model (figure 1) for which they contributed data.

Qualitative data included verbal reports from project staff (#1. #2, #10, #16); semistructured interviews with interventionists and participants in the intervention and control arms of the pilot RCT (#8, #9, #12, #13, #14, #15, #16, #17, #19, #20, #21); minutes of meetings (#3); emails (#4), website development reports (#6); and fidelity assessments (#17). Semistructured interviews, conducted face-to-face, were digitally audio-recorded and transcribed verbatim. The median length of interviews was 30 min (range 11–87) for PWCF, 86 min (63–102) for interventionists and 62 min (51–66) for CF team members.

Quantitative data included implementation log entries and data management reports (#3), questionnaire data derived from secondary clinical outcome measures described in table 1 (#7, #28, #29, #30, #31, #32, #33); an interventionist-completed structured questionnaire on interventionist confidence post-training (#9); structured interventionist fidelity assessments in which audio-recordings of intervention sessions were coded using a fidelity scoring system which assessed whether each component of the intervention was delivered and the quality of that delivery (#11, #17); CFHH click analytics (#13, #14, #15, #18, #20, #21, #22, #23, #24, #25, #26); session frequency and duration records (#15); and adherence data taken from CFHH (#35). Quantitative or descriptive data were collected for the 23 logic model constructs listed in this paragraph as part of the trial protocol, as described in table 1.

## Sampling

Participants were recruited for semistructured interviews. Participants included intervention arm participants (n=14), interventionists (n=3, 0.8 whole-time equivalents at each centre) and members of the wider, multidisciplinary CF team (n=5). Participants were purposively sampled based on site, age, gender, deprivation index, objective and subjective adherence levels (service users) or site and professional category (professionals). Interventionists were interviewed twice—at the beginning and end of the study—patients once. PWCF who consented to be approached for interview were contacted by letter or email and, subsequently, telephone or email depending on preference. Professionals were contacted directly by the study team.

## Data analysis

We conducted a Framework analysis of interview transcripts,[36] within NVivo (QSR International) using multiple frameworks including the Theoretical Domains Framework,[37] a process evaluation framework[38] and the logic model (figure 1).

Using a modified triangulation protocol,[39] we integrated qualitative and quantitative datasets at the programme and the case level.[40] We used a joint display table[41] to summarise datasets for 35 logic model constructs in the Inputs (n=12), Engagement (n=6), Activities (n=7), Immediate outcomes (n=6) and Intermediate outcomes (n=2) columns (figure 1). The fit of data integration was categorised as 'confirmation' (quantitative and qualitative data provided similar findings); 'expansion' (the datasets addressed different or complementary aspects of the phenomenon); or 'discordance' (the datasets were contradictory).[42] We described similar and unique contributions, made by the two datasets, to the research question.[39]

In the 14 intervention participants, for whom both qualitative and quantitative process data were available, we produced case profiles,[43] triangulating qualitative data with individual-participant adherence run charts[44] (online supplemental file 01) and other quantitative process data (see online supplemental file 02 Study protocol, pp29–31). We worked abductively, moving between behaviour change theories[45][46] and contextual observations, agreeing plausible hypotheses to explain patterns which could be tested in future work.[47–50]

We produced a case-ordered descriptive matrix,[51] with cases ranked by average adherence during the last month of the study, to understand how processes and outcomes were mediated by local and individual conditions. Adherence levels of >80% were assessed as high; 50%–80% moderate; <50% low.[14][52] We theorised that high life chaos, as measured by the Confusion, Hubbub and Order Scale (CHAOS)[53] and low motivation would be associated with low adherence. We used four measures to understand motivation: (1) a single item, scored on a 1–7 Likert scale—'I want to do all of my nebuliser treatment' (motivation); (2) a single item, scored on a 1–7 Likert scale, which asked, 'I am confident I can do all of my nebuliser treatments' ('confidence'); (3) the necessities and (4) concerns 5-point subscales of the Beliefs about Medicines Questionnaire nebuliser-specific (BMQ) instrument.[54] Interventionists assessed the participant's motivation to increase adherence on a 1 to 7 scale after discussion with the patient; adequate motivation was necessary before participants could make action plans and do problem-solving activities.

## Approach taken to modifying the intervention

Modifications to the intervention fell into three categories: the software platform; other information technology (IT) infrastructure; and the manual and training. Identified problems and solutions were tabulated following a modified approach of that taken by Bugge et al.[31] Digital platform development was reviewed regularly using the

**Table 1** Quantitative data contributing to the understanding of logic model constructs

| # | Logic model column/construct | Quantitative |
|---|---|---|
| **Inputs** | | |
| 3 | Prescription data | CFHH; problems documented in implementation log. |
| 7 | COM-BMQ questionnaire responses | COM-BMQ, incorporating the Beliefs about Medicines Questionnaire (Nebuliser adherence),[54] one additional belief item, one intention item, one confidence item, and a list of barriers. |
| 9 | Interventionist training programme | Structured questionnaire on interventionist confidence after training programme. |
| 11 | Competency/fidelity assessment | Structured instrument for the assessment of interventionist competence. |
| **Engagement** | | |
| 13 | Clinicians accessing adherence data | CFHH click analytics. |
| 14 | Adherence data tracking | CFHH click analytics. |
| 15 | Participant accessing CFHH | CFHH click analytics. |
| 17 | CFHH Intervention sessions delivered according to manual (fidelity) | Project-specific structured fidelity assessment instrument. |
| 18 | Initial session, and then review at each clinic visit | CFHH click analytics. |
| **Activities** | | |
| *Intervention components for all participants* | | |
| 20 | Self-monitoring adherence | CFHH click analytics. |
| 21 | Tailored education about treatment | CFHH click analytics. |
| 22 | Tailored patient stories (videos) | CFHH click analytics. |
| *Intervention components for those with adequate motivation* | | |
| 23 | Personalised goal-setting | CFHH click analytics. |
| 24 | Goal review | CFHH click analytics. |
| 25 | Personalised action plan | CFHH click analytics. |
| 26 | Tailored problem solving | CFHH click analytics. |
| **Immediate outcomes** | | |
| *For all participants* | | |
| 28 | Acute awareness of adherence/increased Motivation | Subjective adherence single question (self-report estimate of adherence as a percentage); COM-BMQ. |
| 29 | Increased necessity and decreased concern | COM-BMQ and PAM-13.[127] |
| 30 | Increased self-efficacy/motivation | COM-BMQ single question about confidence to adhere; PAM-13. |
| *For those with adequate motivation* | | |
| 31 | Increased self-efficacy/motivation | COM-BMQ single question about confidence to adhere; PAM-13. |
| 32 | Increased habit/reduced chaos | SRBAI automaticity-specific subscale of the Self Report Habit index to capture habit-based behaviour patterns[128]; CHAOS 6-item: measure of life chaos.[53] |
| 33 | Reduced barriers | No change in the group averages for The Beliefs about Medicines Questionnaire—specific (Nebuliser adherence, BMQ 21-item[54]). |
| **Intermediate outcomes** | | |
| 35 | Increased adherence | Nebuliser data (CFHH) |

CFHH, CFHealthHub; CHAOS, Confusion, Hubbub and Order Scale; COM-BMQ, Capability Opportunity Motivation Behaviour Beliefs about Medicines Questionnaire; PAM-13, Patient Activation Measure 13; SRBAI, Self-Report Behavioural Automaticity Index.

'Must have, Should have, Could have, and Won't have but would like' (MoSCoW),[55] often used in agile software development.[56 57]

## Patient and public involvement

Recruitment for the patient and public involvement (PPI) group was achieved by advertising within CF units and on the People in Research website, as well as via group members themselves. Cross-infection between PWCF[58] was prevented by arranging meetings via teleconference. The PPI group gave feedback on intervention data sharing policies, usability and presentation of the website/user guide. In addition, the PPI group piloted the participant information materials and one individual gave feedback on the trial protocol and interview guides (online supplemental file 02).

## Ethics and al approval

The study received approval from London Brent Research Ethics Committee (16/LO/0356). The funder was not involved in the trial design, patient recruitment, data collection, analysis, interpretation, or presentation, writing or editing of the report or the decision to submit for publication. The corresponding author had full access to all the data in the study and had final responsibility for the decision to submit for publication.

## RESULTS

In what follows, we address contextual factors that affected implementation and participant responses, then follow the columns (inputs, engagement, activities, immediate and intermediate outcomes) of the logic model. Online supplemental file 03, tables A–G summarises quantitative process outcomes for 14 case study participants, ranked by objective adherence at the end of the trial. Hashed numbers (#1, #2, etc) indicate cross references to the logic model (figure 1) and supporting evidence in online supplemental fle 04, which summarises data triangulation at the level of individual logic model constructs. Both qualitative and quantitative data were available for 13/34 logic model constructs, providing confirmation of (n=2) or expansion on (n=11) inferences drawn from quantitative data. A case-ordered descriptive matrix based on logic model columns (online supplemental file 05) and run charts annotated with key events (online supplemental file 01) provides an integrated analysis at the level of the participant for 14 'case studies', cross referenced by participant numbers (R02/52, R01/54, etc).

## Contextual factors affecting implementation and participant responses

The key factor affecting implementation was the mixed economy of CF drug delivery systems: the e-Flow (PARI Pharma GmbH, Starnberg, Germany); the iNeb (Philips Healthcare, Eindhoven, Netherlands); and a number of dry powder delivery systems. The e-Flow is the only device able to deliver all the wet nebulised drugs that are used

in CF care. The e-Track we used in this trial was a version of the e-Flow developed to transfer time-stamped and date-stamped data. Most patients at site R01 used e-Flows; switching consenting participants over to the e-Track was generally unproblematic. The e-Flow's competitor, the iNeb, cannot deliver aztreonam and requires double-chamber filling to deliver tobramycin, so it is not suitable for all patients. The data transfer version of the iNeb, the BiNeb, is a prototype for which limited numbers are available. We were unable to secure approval to integrate the BiNeb into CFHH in time to incorporate it into this study. At site R02 where iNebs were commonly used, those who were familiar with and liked the iNeb were less keen to swap to an alternative device; some who swapped to the e-Track, later wanted to move back to the iNeb. A minority of patients use dry powder delivery systems, none of which have data transfer versions. We were unsuccessful in engaging any of the companies producing dry powders in time to get dry powder systems integrated into CFHH, meaning that dry powder users could not be recruited to this feasibility study. Making nebulisers with data recording and transfer capability available within hospitals following local delivery took prolonged engagement with medical engineering departments to obtain local safety approvals. For more than one participant, the strength of their mobile data signal affected 2net Hub connectivity with the central server (Implementation log, 19 October 2016).

Through meetings with site staff, the team identified a range of human factors that also affected implementation, in particular: the availability of out-patient rooms; the need to clean rooms after each consultation for cross-infection control purposes; and the expectation that, during hospital visits, outpatients will see the whole each member of the multidisciplinary team (MDT) separately. The struggle for clinic space and patient convenience resulted in more home visits than anticipated for consent and review meetings, informed by local lone-working policies. Reorganisation of one CF Centre, involving the transfer of patients from the care of one local hospital to another, had created discontent among some patients involved in the trial.

## Inputs

The study chief investigator reported introducing local site investigators, centre directors and MDTs to CFHH (#1). Through case reports, he conveyed that relying on forced expiratory volume in 1 second, symptoms and body mass index for CF management alone is inadequate and that objective adherence data could help overcome the 'lamppost syndrome',[59] also known as the 'streetlight effect'[60 61] or 'drunkard's search' (p11[62])—a type of availability bias.[63] The chief investigator reported feeling that site investigators at both centres were fully bought in, but that one clinician (not an investigator) believed that the disparities between subjective and objective adherence[23] were overstated (#2).

Interventionists entered prescription data into CFHH based on patient records and self-reported treatment regimen (**#3**). Occasionally, interventionists were slow to make monthly prescription checks when prompted by system alerts, resulting in apparent adherence levels of over 100%, traced to the use of alternating treatment regimens[64] (Implementation Log, 1 December 2016, TMG minutes 10 January 2017). Nebulisers with data recording and transfer capability (**#4**), 2net Hubs (**#5**), the CFHH website and mobile application (**#6**), were made available (emails to project manager 20 May 2016, 23 June 2016). The Capability Opportunity Motivation-Beliefs about Medicines Questionnaire (COM-BMQ—see online supplemental file 02)[54] questionnaire data (**#7**) was collected in CFHH (online supplemental file 06, tables 1–22 and figures 1–9).

Interventionists were complimentary about the intervention manual (**#8**) and highly satisfied with training, but suggested that future courses involved a case study approach, following a patient through the intervention to illustrate its different aspects (**#9**, online supplemental file 04). A member of the research team (MH) acted as an intervention mentor to interventionists (**#10**). Interviews (SD) and observations (MH, HC) identified differences in the way site investigators interacted with interventionists, with one giving more intensive practical support, through weekly meetings and problem solving (not prescribed by the intervention), than the other. Fidelity data were collected on all three interventionists and the fidelity assessment instrument was modified before use in the full RCT (**#11**). During interviews, interventionists were enthusiastic about intervention processes (**#12**). As sites struggled to find space or time for consent/intervention encounters in clinic, the study team requested an increase in the number of home visits (Implementation log 19 October 2016). As a result of initial problems in contacting participants and the need for flexibility in arranging meetings out of usual clinic hours, the study team requested flexible working in which the team worked 12:00 to 20:00 2 days a week (interviews & TMG minutes 29 November 2016).

### Engagement

Interviews and click analytics showed that MDT members did not access adherence data (**#13**), aside from in the form of bar charts brought to MDT meetings by interventionists. It is important to note that extending the use of CFHH to the MDT was not an objective of the trial and no training was given in this regard. Click analytics showed that interventionists tracked adherence (**#14**). Of 14 case study participants, 3 did not contribute complete adherence data: R02/42 and R02/02 withdrew, while R02/03 was lost to follow-up. In other participants, flatlines in adherence data caused concern (online supplemental file 01). Flatlines at the beginning of the study (eg, R01/39, R01/48) indicated technical problems with pairing nebulisers and hubs. Flatlines at the end of the study period (eg, R01/42, R01/44, R02/12) were confirmed as the genuine recording of non-adherence through the use of adherence data beyond the end of the study period, interview data, self-report subjective adherence and the Medication Adherence Data-3 (online supplemental file 03, table F).

Click analytics showed the median number of participant CFHH sessions was three (**#15**, online supplemental file 03, table C). Of those with low usage, initial technical problems (R01/02, R01/48) and initial lack of availability of a mobile application (**#6**) were potential contributing factors. Some case study participants showed moderate (R02/52, R01/54 and R01/40: 9–13 sessions) or high use (R02/12 and R01/42:>40 sessions). Push notifications—user-defined messages from the server which give participants congratulations or reminders about adherence behaviour—were not available in the pilot trial (**#16**).

Based on fidelity assessment of intervention session recordings, the *content* fidelity of face-to-face interactions, was excellent (100%)—with all aspects delivered as per the manual (**#17**). Delivery *quality* fidelity was more variable (60%–92%). The generation of goals and action plans was sometimes too directive rather than negotiated and supportive. Interviews demonstrated that assessing the true level of motivation to adhere to treatment was challenging; sometimes those with insufficient intrinsic motivation (eg, R01/48, R01/54 and R02/03) were assessed as having sufficient motivation and inappropriately tasked with setting and reviewing goals, making action plans and problem solving (see below #23–#26). These individuals were variably motivated by wanting to prove themselves to MDT members, who had doubted their adherence (R01/49 and R01/54, online supplemental file 05), or by helping the research:

> I made that special effort 'cause I was taking part in this trial…I didn't see how it was going to make me better. (R01/48)

Interaction with these individuals should have been confined to relationship-building and trust-building. Fidelity assessment of recordings identified that, in interactions with the adequately motivated, the focus was not always on the most active ingredients—goal-setting, action planning (habit formation) and problem solving/coping planning. Participant run charts (online supplemental file 01) revealed a disparity in whether and when review visits happened (**#18**).

### Activities

In interviews, CF team clinicians (as distinct from the interventionists) confirmed they were not monitoring adherence as part of usual care (**#19**). Participant R01/02 complained that the research focus on adherence was '*parallel rather than integrated'* with mainstream clinical management. However, the intervention was designed to be interventionist delivered allowing individual randomisation in a system without contamination of controls rather than an intervention aimed at achieving system change which would have required a cluster trial design.

Participants' clicks (median 11) on the CFHH 'How am I doing?' (run charts) page sometimes related to a limited number of sessions. In interviews, one moderately frequent user (R01/54) only accessed this page to check their data were uploading. Other moderate/frequent users described this page as important for adherence self-monitoring (**#20**), even when their grasp of their own adherence was poor (R01/49).

In interviews with participants, for tailored education about treatment (**#21**), participants accessed particular education pages for specific issues, such as nebuliser malfunction, which was viewed as, 'more down to earth' than technical manuals. In particular a video about the treatment action of Dornase alfa, was often praised, as a means of educating others about CF; 'Talking heads' videos (in these videos people with CF described strategies for successful nebuliser use, **#22**) divided opinion: for some, the opportunity for social comparison[65] provided relief and reassurance; those who were less appreciative were those who found comparisons with people healthier than themselves could make them feel as though they were not doing well and comparisons with those less healthy could make them fearful of the future.

Other activities (**#23–#26** on the logic model) required participants to have adequate levels of motivation. Interventionists classified all but one case study participant (R01/44) as having adequate motivation (online supplemental file 03, table B) and therefore eligible for further tailored intervention. But, as detailed above (see #17 in the engagement section), this was sometimes based on inadequate discussion with the participants. In interviews, participants generally reported setting goals (**#23**), but fidelity assessment showed that goals were sometimes formulated by interventionists rather than by participants (see #17). The mean number of review sessions (**#24**) over 5 months was 1 (online supplemental file 03, table E); this was fewer than intended, likely reflecting a failure of the study team to set appropriate expectations and a lack of time created by the high pace of recruitment (problem log entries: 31 January 2017; 13 February 2017). Two individuals (R01/39 and R01/40) received their first face-to-face session with an interventionist over halfway through the study period (online supplemental file 01). CFHH action plan (**#25**), problem solving and coping plan (**#26**) pages were accessed a median of two, three and one times, respectively (online supplemental file 03, table E). Interviews data suggest action/coping plans were completed during intervention visits but not accessed by participants otherwise. In interviews, some participants said they were reassured by the presence of, and sometimes reported insights from, problem-solving modules, such as what to do when going on holiday. However, the use of action plans was disliked by some participants who found writing down the action plans like '*going back to school*'. This dislike at least partly reflected the generation of action and coping plans by interventionists rather than by the participants themselves (see #17).

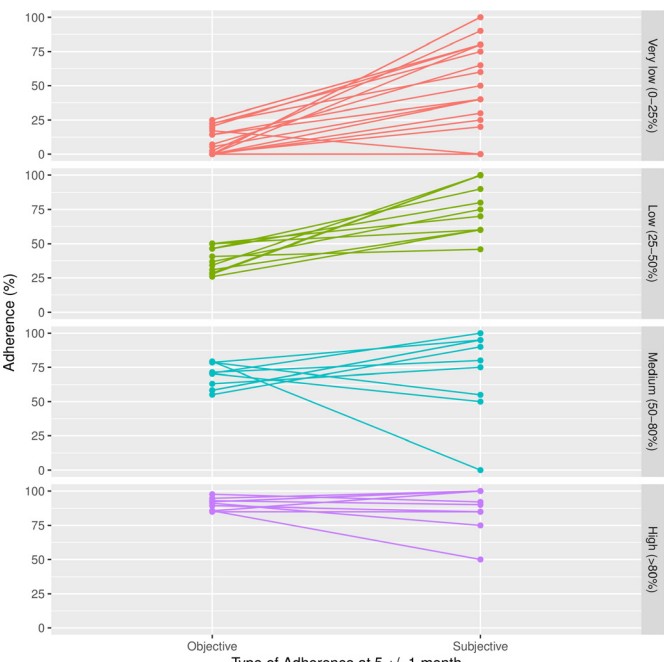

**Figure 3** Objective versus subjective adherence at 5 (±1) months stratified by adherence.

### Immediate outcomes

The pilot was not designed to disseminate the intervention across the centre and with minimal monitoring by professionals within the wider CF team (see #19) routine medical care was not informed by adherence (**#27**). Unsurprisingly, given the lower than expected face-to-face contact (**#18, #24**), intervention arm group averages for immediate (process) outcomes (**#28–33**) changed little over 5 months, with the exception that there was a mean reduction of 1.84 (SD 3.44) barriers to adherence per person (**#33**), which could be the outcome of problem solving and education about treatment processes (online supplemental file 03, table F). Frequent use of CFHH and self-monitoring in particular (see above, #20) did not necessarily mean that self-reported subjective adherence and electronically captured objective adherence were well aligned (**#28**, online supplemental file 03, tables F and G). A post hoc paired comparison of subjective and objective adherence at 5 (±1) months (figure 3) suggests that higher adherers were more uniformly accurate in their understanding of their own adherence, whereas low adherers could be overly optimistic.

### Intermediate outcomes

Item **#34** of the logic model, treatment optimisation, is defined by NICE as, 'a person-centred approach to safe and effective medicines use' to ensure best outcomes.[66] Treatment optimisation is a service-level objective, which was beyond the scope of our patient-focused intervention but is the subject of related ongoing research (see Discussion section). During interviews, RCT participants in the intervention arm described behaviours that would affect treatment optimisation, for instance taking holidays from their treatment. Levels of CF treatment adherence (**#35**)

were 10% (95% CI: −5.2 to 25.2) higher in the intervention arm (online supplemental file 06). We developed a number of theories about why some intervention patients did or did not increase their adherence (**#35**) during the analysis. In some cases the run charts illustrated, inline with Control Theory, the goal-directed nature of behaviour and how it is regulated by feedback control processes.[67] For example, R01/39 and R01/49 seemed to show improvement shortly before planned face-to-face visits from interventionists (online supplemental file 01). R01/39, who seemed intrinsically motivated when interviewed, sustained improvement in adherence beyond the trial period through what they described as positive interaction with the interventionist. Others, who seemed more extrinsically motivated in interviews (R01/49, R01/54, R01/48: see #17), did not sustain adherence, with charts suggesting an effortful, 'all-or-nothing' pattern. At baseline, R02/07 had no well-established routine (CHAOS score of 10: online supplemental file 01), implying substantial self-regulatory effort to achieve higher adherence. In their interview, this participant reported finding habit formation parts, such as goal-setting, helpful which may have enabled him to maintain high adherence with reduced effort, as measured by increased habit and reduced life chaos and barriers (change scores −5 and −3, respectively: online supplemental file 03, table F). Finally, it is important to understand that individual-level adherence can be unstable over time (online supplemental file 01, see especially, R01/54, R01/48) highlighting the problem of assessing adherence as a 'snapshot' in a pretest/post-test analysis, rather than in a continuous assessment over time.

Several participants with low baseline adherence appeared to have responded well to the intervention. R01/40 had high motivation (online supplemental files 01 and 03, table B), possibly due to the salience of a recent hospitalisation for IVAB treatment of an exacerbation. Click analytic data showed high engagement, with independent access of the website and use of problem-solving tools. However, in other patients, case study run charts (online supplemental file 01) showed that measuring change in average objective adherence between baseline and 5 months sometimes masked periods of success in between (eg, R01/02, and R02/12). Without looking at adherence graphs, and only measuring objective adherence at baseline and 5 months, this would have been missed (see Discussion section). Interview data offered some reasons for improved adherence. While R01/49 had not made an action plan and their subjective adherence was optimistic (online supplemental file 03, table F), their objective adherence increased from low to moderate over the trial period (online supplemental file 01); their motivation also increased and self-reported barriers decreased (online supplemental file 03, table F), potentially through their high use of problem-solving modules and self-monitoring (online supplemental file 03, table D). R01/02's run chart also showed a period of improvement, ending after the last review visit (online

supplemental file 01); nonetheless, reduced life chaos (online supplemental file 03, Table F) and interview data suggested an established routine and reduced barriers associated with intensive face-to-face therapist interaction and action/coping plans (online supplemental file 03, table D). The tailing off of adherence after the end of the trial in some case study participants may indicate that adherence remained effortful or participation in the trial was motivated by altruism not help-seeking (see quotation from R01/48, above).

### Modifications to the intervention

Online supplemental file 07 documents 14 technical changes that will be made for the full-scale RCT, based on the process evaluation findings, to CFHH (n=5), IT infrastructure (n=1) and to the interventionist training, manual and procedures (n=8). To prevent adherence data flatlines, nebulisers (**#4**) and 2net Hubs (**#5**) will be paired at the factory. Three changes to CFHH (**#6**) will make it easier for interventionists to view/edit prescription data and to handle alternating treatment regimens (**#3**). Other changes to CFHH will include making graphs more easily interpretable and, based on interview data and PPI feedback, adding descriptions to videos. Changes to the interventionist manual (**#8**) will increase the emphasis on 'active ingredients', introduce intervention triggers for reduced adherence or exacerbations and introduce new habit formation sessions. The need for increased numbers of protocolised intervention review sessions arose because, in the feasibility study, a focus on RCT recruitment targets gave interventionists inadequate time to deliver review visits (**#18, #24**), critical for updating personalised action plans (**#25**) and updating coping plans (**#26**). Training (**#9**) in the full-scale trial will be delivered as an intensive 1-week course, with more explicit focus on intervention fidelity, supported by new case study data and role plays to ensure baseline competency (**#17**).

### DISCUSSION

The process evaluation identified elements of the intervention which could be improved and 14 changes were documented. The complex intervention was developed using mixed-methods research with an interdisciplinary, person-centred and iterative approach.[68–74] The mere usage of a digital behaviour change intervention may not indicate engagement or lead to desired outcomes;[68 73 75–78] there is no simple dose–response relationship.[79] In fact, for those with low motivation and low confidence, evidence of non-adherence can be threatening.[80 81] With different baseline motivation and life chaos, a population-level definition of 'effective engagement'[70] may be infeasible, but contextual and motivational data may still explain patterns observed in run charts.[82] What may matter more than defining engagement is the correct assessment and tailoring of management to different psychosocial barriers.[69 83–91] Our study suggests that digital systems

cannot replace, only complement, face-to-face interaction between health professionals and patients,[92–95] potentially creating a sense of 'accountability' consistent with control theory.[46 96] However, it is important to recognise that in the absence of objective adherence data, clinicians and patients will find it difficult to even begin to engage with behaviour change.

Chronic disease self-management is a complex and multifactorial problem and, we were unable to cover all of the analyses that many would consider relevant. For instance, although the intervention is meant to increase health literacy through education, we cannot rule out that baseline socioeconomic status, known to affect health literacy, outcomes and self-management,[97–99] was not a factor. Another limitation of this study is that we interviewed just over only one quarter of the pilot trial sample. Given a relatively homogeneous population, narrow, exploratory study aims and the use of established theory, 14 interviews should be adequate to discern common perceptions and experiences.[100 101] In the full-scale evaluation of this intervention (see last paragraph of this section), the process evaluation will involve a user acceptability survey of ~250 intervention users from 19 centres and face-to-face interviews with over 50 intervention users, interventionists and clinicians. As in many other process evaluations, we will use maximum variation sampling on sociodemographic characteristics and baseline adherence, alongside triangulation, to minimise the risk of bias.[102] Additionally, readers should be aware that small-scale feasibility work does not generalise in every regard when scaled up in larger scale studies.[103 104] Finally, early health economic modelling of the cost-effectiveness,[105] was not updated as part of this feasibility work, but will be revisited in 2021 as part of the full-scale evaluation.

Our use of objective adherence measurement overcomes the limitations of previous studies[106] and confirms that subjective and objective adherence are poorly aligned.[23] This process evaluation has succeeded in demonstrating that delivery of this intervention is possible in busy clinical settings; participant uptake was high and, with further development on the basis of these findings, the process of gathering objective adherence data and implementing it alongside a behavioural intervention is both possible and effective.

Given the known difficulties with nebuliser use among PWCF, interventions that can make it less effortful are important.[107] In particular, healthy behaviours are better predicted by a patient's level of automatic behavioural repetition than their beliefs or experiences, meaning a focus on increasing habit strength is critical for chronic disease self-management.[108] Through delivery of intervention components designed to promote habit formation, we intend to reduce effort with the CFHH intervention. We are limited in drawing conclusions as to the impact of habit formation components of the intervention from this analysis; this is mostly due to the limited time constraints of the feasibility study leaving insufficient opportunity for habit formation.[109] However, there was some indication that habit components were useful and we have elsewhere demonstrated the importance of habit in high adherence.[110 111] It has also been indicated that adherence interventions focusing on habit formation are the most effective.[112]

Successful habit formation will reduce burden by making sustained self-care automatic. The CFHH intervention aims to deliver the fall in burden highlighted by the Lind alliance prioritisation exercise as the most important goal of CF research.

To date, there is little previous research showing the effects of giving patients access to their data, with respect to health outcomes and cost-effectiveness. Amidst the evidence that does exist, the research is generally poor and lacks information about context and implementation.[113 114] Following modifications made to our complex intervention, the full-scale RCT across 19 UK centres (ISRCTN55504164) will provide high-quality evidence, indicating the impact of adherence data on sustained self-care. The full-scale RCT will include a further process evaluation and health-economic modelling. Furthermore, the CFHH Data Observatory (ISRCTN14464661) following on from the RCT will address the issue of how to embed the use of adherence data in routine practice for healthcare professionals.[115–119] The sites involved in the reported pilot study have now transitioned into the Data Observatory, eventually to be joined by sites involved in the full-scale RCT. Data collected in the Data Observatory quality improvement project will be used in the development of generalisable theory and practical guidance about the collaborative use of adherence data,[120–122] with a focus on optimising the use of healthcare resources and improving patient care.[66 123] The Observatory will act as a platform for efficient trials,[124 125] providing an opportunity to share processes and improvement activities to enable participating CF clinical research teams to meet the demands of future research.[126]

## CONCLUSIONS

We have developed a theory-based complex intervention to help PWCF adhere to their medication and form habits of sustained self-care. The process evaluation identified potential sources of intervention failure and modifications have been made accordingly. With improved intervention processes, it is feasible and acceptable to support sustained self-care via medication adherence through the application of behaviour change theory delivered through digital and human components.

**Author affiliations**
[1]Clincal Trials Research Unit, University of Sheffield, Sheffield, UK
[2]School of Health and Related Research, University of Sheffield, Sheffield, UK
[3]Centre for Behavioural Science and Applied Psychology, Sheffield Hallam University, Sheffield, UK
[4]Department of Health Sciences, University of York, York, UK
[5]Sheffield Adult Cystic Fibrosis Unit, Sheffield Teaching Hospitals NHS Foundation Trust, Northern General Hospital, Sheffield, UK

[6] Wellcome-Wolfson Institute For Experimental Medicine, School of Medicine, Dentistry and Biomedical Sciences, Queen's University Belfast, Belfast, UK
[7] Wessex Adult Cystic Fibrosis Service, University Hospital Southampton NHS Foundation Trust, Southampton, UK
[8] Wolfson Cystic Fibrosis Centre, Nottingham University Hospitals NHS Trust, Nottingham, UK
[9] Health eResearch Centre - Division of Imaging, Informatics and Data Sciences, School of Health Sciences, Faculty of Biology, Medicine and Health, The University of Manchester, Manchester Academic Health Science Centre, Manchester, UK

**Acknowledgements** We would like to thank the trial participants who offered us their time. We gratefully acknowledge the clerical and other input of Helen Wakefield, Heather Dakin, Katie Shore and Louise Turner, trial support officers. We would also like to thank Claire Oliver, Fiona Haynes and Lisa Evans for their commitment and hard work in delivering the intervention and supporting the study. SJW is a Senior Investigator in National Institute for Health Research (NIHR).

**Contributors** DH (Assistant Director, CTRU), SJD (Research Associate) and LM (Statistician), together produced the first draft of the report. The following authors conceived of or designed the work: MJW (Consultant Respiratory Physician), AO (Professor of Health Services Research), SJW (Professor of Medical Statistics and Clinical Trials), MAA (Professor of Health Psychology), MH (Physiotherapist), JB (Professor of Physiotherapy), JA (Professor of Health Informatics), DB (Patient and Public Involvement Representative) and DH. The following authors were involved in the acquisition of data for the work: JN (Consultant in Respiratory Medicine), MIA (Consultant in Respiratory Medicine), JD (Consultant in Respiratory Medicine), SJD, MAA, CM, HC, AS (Research Associate), SK (Research Associate), MH and PW (mHealth Applications Manager). The following authors were involved in the analysis of data: DH, SJD, AO, DB, AS, SK, MAA, LM, SW (Data Manager). DH, SJD, MAA, LM, SW, CM, HC, LR, MH, JB, JN, MIA, JD, DB, SJW, AO and MJW were involved in the interpretation of data for the work. All authors were involved in the final approval of the version to be published. All authors agree to be accountable for all aspects of the work in ensuring that questions related to the accuracy or integrity of any part of the work are appropriately investigated and resolved.

**Funding** This report presents independent research funded by the NIHR under its Grants for Applied Research Programme (Grant Reference Number RP-PG-1212–20015).

**Disclaimer** The views expressed in this article are those of the author(s) and not necessarily those of the NHS, the NIHR, or the Department of Health and Social Care.

**Competing interests** MJW received funding from Zambon and support from Philips Respironics for the early intervention development work. This has not had any direct influence on the feasibility study reported here. In addition, MJW has worked with Pari to carry out studies using the e-track. This has not had any direct influence on the feasibility study reported here. The University of Manchester software team received funding from Pari to create a medication reporting component within the CFHealthHub software. This has not had any direct influence on the feasibility study reported here.

**Patient consent for publication** Not required.

**Ethics approval** The study received ethical approval from the London Brent Research Ethics Committee (16/LO/0356).

**Provenance and peer review** Not commissioned; externally peer reviewed.

**Data availability statement** No data are available. Requests for further data not available in this publication can be directed at Sheffield Clinical Trials Research Unit. Email: ctru@sheffield.ac.uk Tel: 0114 222 0866.

**ORCID iDs**
Daniel Hind http://orcid.org/0000-0002-6409-4793
Alex Scott http://orcid.org/0000-0001-7426-7099
Stephen J Walters http://orcid.org/0000-0001-9000-8126
Alicia O'Cathain http://orcid.org/0000-0003-4033-506X

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
