## [Reviewer comments · BMJ Open]

ARTICLE DETAILS

TITLE (PROVISIONAL)	Feasibility study for supporting medication adherence for adults with cystic fibrosis: mixed-methods process evaluation
AUTHORS	Hind, Daniel; Drabble, Sarah; Arden, Madelynne; Mandefield, Laura; Waterhouse, Simon; Maguire, Chin; Cantrill, Hannah; Robinson, Louisa; Beever, Daniel; Scott, Alex; Keating, Sam; Hutchings, Marlene; Bradley, Judy; Nightingale, Julia; Allenby, Mark; Dewar, Jane; Whelan, Pauline; Ainsworth, John; Walters, Stephen; Wildman, Martin; O'Cathain, Alicia

VERSION 1 – REVIEW

REVIEWER	M Lopes-Pacheco Faculty of Sciences of the University of Lisbon, Portugal
REVIEW RETURNED	10-May-2020

GENERAL COMMENTS	In the manuscript “Feasibility study for supporting medication adherence for adults with cystic fibrosis: mixed-methods process evaluation”, Hind and collaborators tested a complex mix of methods to assess the adherence of people with CF (PwCF) to nebulizer therapies. This is an important concept that certainly needs the attention of CF researchers as current adherence for PwCF to therapies is not optimal. Nevertheless, there are several limitations in this study (also pointed out by the authors). Specific comments: 1) A major limitation of this study is the very small sample size of 14 participants with two withdraw and one loss of follow-up.2) “The struggle for clinical space and patient convenience...” – Are the authors able to provide information regarding the socioeconomic status and/or work type of each participant? This could provide some clues to identify the issues in achieving optimal adherence.3) “...higher adherers were more uniformly accurate in their understanding of their own adherence...” – This could also be due to a better socioeconomic status.4) The development of educational approaches may help in establishing a routinely and sustained self-care in order to achieve the greatest therapeutic outcomes. This should be acknowledged.5) “...long-term inherited condition...” (page 8, line 106) – This expression sounds awkward. Please replace “life-threatening inherited disorder/disease” or similar.6) “...80,000 people worldwide...” – More specifically over 90,000 people currently.
--

REVIEWER	Marie Viprey Hospices Civils de Lyon, France
REVIEW RETURNED	29-May-2020

GENERAL COMMENTS	Thank you for your manuscript presenting the results of a well conducted process evaluation of an adherence support intervention for people with cystic fibrosis. The article is clear, well-written and relevant to the scientific knowledge. It has a well-defined method. I have therefore some comments on your manuscript: Background: 1) Page 6 line 127 “Consistent with identified research priorities[25,26]”: the research priorities from which the developed intervention stems could be briefly recalled in the sentence in order to make the sentence more explicit, and specify who set these priorities. 2) Page 6 lines 129-130: it would be useful for the reader's understanding to clearly specify the objective of the RCT pilot. Methods: 3) Page 8 line 177 “Health professionals (n =3) included a clinical psychologist, a physiotherapist and a social worker.”: does the term "health professionals" refer to interventionists? If yes, it would be more appropriate to use the term “interventionist” rather than the term "health professionals" which has not been used before in the paragraph describing the intervention. 4) Page 10 lines 222-223 “These 23 logic model constructs were collected as part of the trial protocol from sources described in Table 1.”: What does the number 23 stand for? 5) Page 10 lines 224-227: This paragraph on semi-structured interviews could be moved just after the paragraph on qualitative data (line 212). 6) Page 12, line 283: the abbreviation “IT” must be described. Results: 7) The results are very rich, even too dense. Some details, such as email references or implementation log references, could be removed for easier reading. 8) Page 15 line 357: the abbreviation “MDT” must be described. Discussion: 9) The limitations of the study should be presented and discussed.
---

REVIEWER	Melissa Basile Northwell Health USA
REVIEW RETURNED	13-Jul-2020

GENERAL COMMENTS	Abstract: Outcome measures – only feasibility and acceptability, and fidelity-were there others? Did they measure costs? Strengths and limitations – blurred, should clearly state the strengths and the limitations. Main text: Very important for CF adherence related health outcomes research. Currently very difficult to measure adherence objectively that is cost effective. Background – good introduction as to why it is critical to address poor adherence in CF, and why it is important to find objective measure for tracking adherence. This section should also provide a discussion about other objective measures, and some of the pros and cons of these other measures. One area to explore more fully in the literature review is in the area of feasibility of implementation – the extent to which this intervention is very difficult for clinicians to adopt. Methods –
--

	Description of the complex intervention - the authors spend too much time describing the complex intervention, making it difficult to remain focused on the actual objectives of the manuscript – the whole first section of the methods (roughly 2 pages) is spent on this wider study (which is already published and the protocol is included as a supplemental). While this provides a good amount of transparency, it ultimately distracts the reader from overall purpose of this manuscript, which is a process evaluation. To a large extent, it's unclear what value this process evaluation adds to the feasibility study already undertaken. Overall, the study team did a very good job in terms of study development, and data transparency for the wider study, and ultimately there is a need for process evaluations for feasibility studies such as this. If the authors could put greater emphasis on the process evaluation, this is a study worth publishing.
--	--

VERSION 1 – AUTHOR RESPONSE

Reviewer: 1

Reviewer Name: M Lopes-Pacheco

Specific comments:

1) A major limitation of this study is the very small sample size of 14 participants with two withdraw and one loss of follow-up.

We have added the following section to the discussion (lines 618-628):

"Another limitation of this study is that we interviewed just over only one quarter of the pilot trial sample. Given a relatively homogeneous population, narrow, exploratory study aims and the use of established theory, fourteen interviews should be adequate to discern common perceptions and experiences[100,101]. In the full-scale evaluation of this intervention (see below), the process evaluation will involve a user acceptability survey of ~250 intervention users from 19 centres and face-to-face interviews with over 50 intervention users, interventionists and clinicians. As in many other process evaluations, we will use maximum variation sampling on sociodemographic characteristics and baseline adherence, alongside triangulation, to minimise the risk of bias[102]."

2) "The struggle for clinical space and patient convenience..." – Are the authors able to provide information regarding the socioeconomic status and/or work type of each participant? This could provide some clues to identify the issues in achieving optimal adherence.

We thank the peer reviewer for prompting us to clarify this point, which was not related to socio-economic status. We have added the following (lines 360-364):

"Through meetings with site staff, the team identified a range of human factors that also affected implementation. , in particular: the availability of out-patient rooms; the need to clean rooms after each consultation for cross-infection control purposes; and, the expectation that, during hospital visits, outpatients will see the whole each member of the multidisciplinary team separately."

3) "...higher adherers were more uniformly accurate in their understanding of their own adherence..." – This could also be due to a better socioeconomic status.

We agree that this is a possibility. As our statistician has moved to another institution and no longer has access to the data set, we have chosen to deal with this as a limitation of the study (lines 614-8):

"Chronic disease self-management is a complex and multi-factorial problem and, we were unable to cover all of the analyses that many would consider relevant. For instance, although the intervention is meant to increase health literacy through education, we cannot rule out that baseline socio-economic status, known to affect health literacy, outcomes and self-management[94–96], was not a factor."

4) The development of educational approaches may help in establishing a routinely and sustained self-care in order to achieve the greatest therapeutic outcomes. This should be acknowledged.

We agree. Indeed, we did highlight that CFHH has educational functions (line 176), but have taken the opportunity to signpost the reader to more information about how this was developed (citation 33) and have added the following detail (lines 176-182):

"Educational modules within CFHH include: 'What is Cystic Fibrosis?'; 'What does my IV treatment do?'; 'I'm not convinced that my nebuliser treatment works'; 'What does my nebuliser treatment do and why should I take it?'; 'Why is it important that I do my nebuliser treatment every day?'; and, 'I have concerns about my nebuliser treatments'. The nebuliser medication information displayed to the user in these sections are tailored to them based on a baseline assessment of motivation, so as not to overwhelm them."

5) "...long-term inherited condition..." (page 8, line 106) – This expression sounds awkward. Please replace "life-threatening inherited disorder/disease" or similar.

We have replaced long-term with life-threatening.

6) "...80,000 people worldwide..." – More specifically over 90,000 people currently. Thank you. Text changed and citation updated.

Reviewer: 2

Reviewer Name: Marie Viprey

1) Page 6 line 127 "Consistent with identified research priorities[25,26]": the research priorities from which the developed intervention stems could be briefly recalled in the sentence in order to make the sentence more explicit, and specify who set these priorities.

Thank you. We have extended the sentence as follows: "Reducing treatment burden has been identified as a key research priority for PWCF and clinicians by both the Cystic Fibrosis Foundation and the James Lind Alliance[25,26]."

2) Page 6 lines 129-130: it would be useful for the reader's understanding to clearly specify the objective of the RCT pilot.

The sentence now reads: "This article presents the results of a process evaluation that was undertaken alongside a pilot RCT, the objectives of which were to determine the feasibility of a full-scale RCT[27]."

3) Page 8 line 177 "Health professionals (n =3) included a clinical psychologist, a physiotherapist and a social worker.": does the term "health professionals" refer to interventionists? If yes, it would be more appropriate to use the term "interventionist" rather than the term "health professionals" which has not been used before in the paragraph describing the intervention.

We agree. Change made.

4) Page 10 lines 222-223 "These 23 logic model constructs were collected as part of the trial protocol from sources described in Table 1.": What does the number 23 stand for?

We apologise that we have not made ourselves clear. The sentence now reads:

"Quantitative or descriptive data was collected for the 23 logic model constructs listed in this paragraph as part of the trial protocol, as described in Table 1."

5) Page 10 lines 224-227: This paragraph on semi-structured interviews could be moved just after the paragraph on qualitative data (line 212).

Change made.

6) Page 12, line 283: the abbreviation "IT" must be described.

Abbreviation clarified and also added to the list of abbreviations at base of article.

7) The results are very rich, even too dense. Some details, such as email references or implementation log references, could be removed for easier reading.

We appreciate the peer reviewer's request. Our experience is that other readers would be concerned at a lack of accountability if we did not source claims in this way.

8) Page 15 line 357: the abbreviation "MDT" must be described.

Abbreviation described.

9) The limitations of the study should be presented and discussed.

In addition to the response to sample size concerns (please see above), we have added the following sentence which addresses the generalisability of small-scale feasibility work at a limited number of centres (lines 628-9):

"Additionally, readers should be aware that small scale feasibility work does not generalise in every regard when scaled up in larger scale studies[94,95]."

Reviewer: 3

Reviewer Name: Melissa Basile

Abstract:

Outcome measures – only feasibility and acceptability, and fidelity- were there others?

As the methods detail, an enormous battery of quantitative assessments and theoretical constructs were investigated. In the context of a 250-word abstract, we are constrained to discuss the key, higher level objectives.

Did they measure costs?

Clarification added to limitations paragraph (lines 630-2):

"Finally, early health economic modelling of the cost-effectiveness[96], was not updated as part of this feasibility work, but will be revisited in 2021 as part of the full-scale evaluation."

Strengths and limitations – blurred, should clearly state the strengths and the limitations.

As part of our response to all three reviewers, we have developed a limitations paragraph.

Main text:

Very important for CF adherence related health outcomes research. Currently very difficult to measure adherence objectively that is cost effective.

We thank the reviewer for recognising the importance of the work. As detailed above, we have flagged ongoing cost-effectiveness work in this area.

Background – good introduction as to why it is critical to address poor adherence in CF, and why it is important to find objective measure for tracking adherence. This section should also provide a discussion about other objective measures, and some of the pros and cons of these other measures.

We have added the following sentences to the Introduction (lines 126-9).

"Hitherto, the most objective surrogate measure of adherence has been the medicines possession ratio (MPR). However, based on the experience of a CF service in Leeds UK, MPR rates of 63%[25] considerably over-estimate adherence compared with nebuliser download data of 36%[26]."

One area to explore more fully in the literature review is in the area of feasibility of implementation – the extent to which this intervention is very difficult for clinicians to adopt.

We agree with the statement but feel it is more of a Discussion point (in that it follows from the problem at hand) rather than an Introduction point (it is not an motivating issue). Indeed, we have already flagged the issue in the Discussion (lines 670-3)

Methods –

Description of the complex intervention - the authors spend too much time describing the complex intervention, making it difficult to remain focused on the actual objectives of the manuscript – the whole first section of the methods (roughly 2 pages) is spent on this wider study (which is already published and the protocol is included as a supplemental). While this provides a good amount of transparency, it ultimately distracts the reader from overall purpose of this manuscript, which is a process evaluation.

The pilot trial paper describes a set of procedures over which the intervention was developed. This paper, necessarily describes the intervention in different terms, in terms of the "long and complex causal pathways" (see MRC Framework on Complex Interventions) by which we believe the intervention works. This type of description is integral to process evaluations (see Moore GF, et al. BMJ 2015;350:h1258–h1258, MRC guidance on which one of our co-authors is also a co-author).

To a large extent, it's unclear what value this process evaluation adds to the feasibility study already undertaken.

The published pilot trial article assesses the feasibility of a research protocol, whereas the current article assesses the feasibility of an intervention.

Overall, the study team did a very good job in terms of study development, and data transparency for the wider study, and ultimately there is a need for process evaluations for feasibility studies such as this. If the authors could put greater emphasis on the process evaluation, this is a study worth publishing.

We would like to reassure the reviewer that every thing in this paper is devoted to identifying and understanding potential barriers to intervention implementation and success (see Objectives), which is the principle function of process evaluations (see Moore GF, et al. BMJ 2015;350:h1258–h1258).

VERSION 2 – REVIEW

REVIEWER	Marie Viprey Hospices Civils de Lyon, France
REVIEW RETURNED	24-Aug-2020

GENERAL COMMENTS	Appropriate responses were provided by the authors to the comments.
---

REVIEWER	Melissa Basile Northwell Health United States
REVIEW RETURNED	29-Jul-2020

GENERAL COMMENTS	My concerns have been addressed in the revision and in the response.
--